# What Deep Representations Should We Learn? – A Neural Collapse Perspective

## Abstract

For classification tasks, when sufficiently large networks are trained until convergence, an intriguing phenomenon has recently been discovered in the last-layer classifiers, and features termed neural collapse (NC): (i) the within-class variability of the features collapses to zero, and (ii) the between-class feature means are maximally and equally separated. Despite of recent endeavors to understand why NC happens, a fundamental question remains: whether NC is a blessing or a curse for deep learning? In this work, we investigate the problem under the setting of transfer learning that we pretrain a model on a large dataset and transfer it to downstream tasks. Through various experiments, our findings on NC are two-fold: (i) when pre-training models, preventing intra-class variability collapse (to a certain extent) better preserve the structures of data, and leads to better model transferability; (ii) when fine-tuning models on downstream tasks, obtaining features with more NC on downstream data results in better test accuracy on the given task. Our findings based upon NC not only explain many widely used heuristics in model pretraining (e.g., data augmentation, projection head, self-supervised learning), but also leads to more efficient and principled transfer learning method on downstream tasks.

## 1 Introduction

Recently, an intriguing phenomenon has been discovered in terms of learned deep representations, in which the last-layer features and classifiers *collapse* to simple but elegant mathematical structures on the training data: (*i*) for each class, the intra-class variability of last-layer features collapses to zero, and (*ii*) the between-class class means and the last-layer classifiers all collapse to the vertices of a Simplex Equiangular Tight Frame (ETF) up to scaling. This phenomenon, termed *Neural Collapse* ($\mathcal{NC}$) (Papyan et al., 2020; Han et al., 2022), has been empirically demonstrated to persist across a variety of network architectures and datasets. Theoretically, more recent works (Fang et al., 2021; Zhu et al., 2021; Zhou et al., 2022; Tirer & Bruna, 2022) justified the prevalence of $\mathcal{NC}$ under simplified unconstrained feature models across a variety of training losses and problem formulations.

Despite of recent endeavors of demystifying such an interesting phenomenon, a fundamental question lingers: *is $\mathcal{NC}$ a blessing or a curse for deep representation learning?* Understanding such a question could address many important but mysterious aspects of deep representation learning. For example, quite a few recent works (Papyan et al., 2020; Galanti et al., 2022; Hui et al., 2022) studied the connection between $\mathcal{NC}$ and generalization of overparameterized deep networks. In this work, we aim to understand transfer learning by studying the relationship between $\mathcal{NC}$ and the transferability of pretrained deep models. Transfer learning has become an increasingly popular approach in computer vision, medical imaging, and natural language processing (Zhuang et al., 2020). With domain similarity, a pretrained large model on upstream datasets is reused as a starting point for fine-tuning a new model on a much smaller downstream task (Zhuang et al., 2020). The pretrained model reuse with fine-tuning significantly reduces the computational cost, and achieves superior performances on problems with limited training datasets.

However, without principled guidance, the underlying mechanism of transfer learning is not very well understood. First, when we are pretraining deep models on the upstream dataset, we lack good metrics for measuring the quality of the learned model or representation. In the past, people tended to rely empirically on controversial metrics for predicting the transferred test performance, such as the validation accuracy on the pretrained data (e.g., validation accuracy on ImageNet (Kornblith et al., 2019)). For example, some popular approaches (e.g., label smoothing (Szegedy et al., 2016)

and dropout (Srivastava et al., 2014)) for boosting ImageNet validation accuracy turn out to hurt transfer performance on downstream tasks (Kornblith et al., 2021). Additionally, when pretraining deep models, many methods improve transferability, such as the design of loss functions, data augmentations, increased model size, and projection head layers (Chen et al., 2020; Khosla et al., 2020), are designed largely based upon trial-and-error without much insight of the underlying mechanism. Second, given the pretrained models, how to efficiently fine-tune the model on downstream tasks remains an open question. Although fully fine-tuning all the parameters of the pretrained model achieve the best performance, it becomes increasingly expensive as the model size grows (e.g., GPT-3 and transformer (Brown et al., 2020; Vaswani et al., 2017; Devlin et al., 2019; Dosovitskiy et al., 2021)). All these challenges call for a deeper understanding of what makes pretrained deep models more transferable.

**Contributions of this work.** In this work, we provide a comprehensive investigation of the relationship between the transferability of pretrained models and $\mathcal{NC}$. As $\mathcal{NC}$ implies that intra-class variability for each class collapses to zero, the representations learned via *vanilla* supervised learning fails to capture the intrinsic dimensionality of the input data, and hence they often result in poor performance of transferability. Intuitively, to make the pretrained models transferable, for each class the learned features should be discriminative but *diverse enough* that they can preserve the intrinsic structures of the input data. On the other hand, on the downstream task when we fine-tune pretrained models, we desire more collapse of the features on the downstream training data.

Based upon such intuitions, we adapt the metrics for evaluating $\mathcal{NC}$ to measure the quality of learned representations in terms of both *intra-class diversity* and *between-class discrimination*. As such, not only can we demystify several heuristics that are widely used in transfer learning, but it also opens a door for designing methods to transfer large pretrained models more effectively. In words, our experimental findings based upon the $\mathcal{NC}$ metrics can be summarized as follows.

- **The transferability of pretrained models correlates with learned feature diversity on the source dataset.** By evaluating the $\mathcal{NC}$ metrics on different loss functions (Hui & Belkin, 2020) and several widely used techniques in transfer learning (e.g., the addition of projection head, different data augmentations (Chen et al., 2020; Chen & He, 2021; Khosla et al., 2020) and adversarial training (Salman et al., 2020; Deng et al., 2021)), we find that to a certain extent,[1] the more diverse the features are, the better the transferability of the pretrained model. This helps to explain the underlying mechanism of many popular heuristics for transfer learning.
- **More collapse of fine-tuned models leads to better test performance on downstream tasks.** In contrast, when we are evaluating different pretrained models on downstream tasks, we observe that more collapsed features on downstream data usually lead to better transfer accuracy. This phenomenon not only happens on the penultimate layer across different pretrained models, but also across different layers of the same pretrained model.
- **pretrained models can be more effectively transfered through $\mathcal{NC}$.** Efficient and effective transfer learning is of paramount importance for large models nowadays. Inspired by the above findings, with the aim to collapse the features of the penultimate layer, we improve the transfer effectiveness while maintaining efficiency by only tuning one additional layer along with an add-on skip connection. We demonstrate that such a transfer learning strategy achieves better performances compared with the traditional fixed feature transfer learning setting and on-par or superior performances compared with full model fine-tuning setting.

**Relationship to prior arts.** The prevalence of $\mathcal{NC}$ phenomenon has caught significant attention both in practice and theory recently, and our work draws the connection between $\mathcal{NC}$ and transfer learning. On the other hand, a few recent works are investigating the properties of deep representations for transfer learning, which is also related to ours. We'd like to summarize and briefly discuss those results below.

- **Understandings of the $\mathcal{NC}$ phenomenon.** There are a line of recent works deciphering training, generalization, and transferability of deep networks in terms of $\mathcal{NC}$, that are related to ours (see a recent review work (Kothapalli et al., 2022)). For training, recent works showed that $\mathcal{NC}$ happens for a variety of loss function and formulations, such as cross-entropy (CE) (Papyan et al., 2020;

---

[1]We find that there is a certain threshold, that the transferability increases with the feature diversity below the threshold but decreases or become uncorrelated beyond it. Increasing the feature diversity will decrease the margin upon the threshold and hence the relationship with transferability becomes more involved with too large feature diversity.

lu2, 2022; Zhu et al., 2021; Fang et al., 2021; Ji et al., 2022; Yaras et al., 2022), mean-squared error (MSE) (Mixon et al., 2020; Han et al., 2022; Zhou et al., 2022; Tirer & Bruna, 2022; Rangamani & Banburski-Fahey, 2022), and supervised contrastive (SupCon) loss (Graf et al., 2021). For generalization, the work (Galanti et al., 2022) shows that $\mathcal{NC}$ also happens on test data drawn from the same distribution asymptotically, but not for finite samples (Hui et al., 2022). Other works (Hui et al., 2022; Papyan, 2020) demonstrated that the variability collapse of features is actually happening progressively from shallow to deep layers, and Ben-Shaul & Dekel (2022) showed that test performance can be improved when enforcing variability collapse on features of intermediate layers. The works Xie et al. (2022); Yang et al. (2022); Thrampoulidis et al. (2022) studied problems with imbalanced training data, showing that fixing the classifier as simplex ETFs improves test performance on imbalanced training data and long-tailed classification problems. For transferability, the work Kornblith et al. (2021) implicitly showed that there is a tradeoff between variability collapse and transfer accuracy by experiments on a variety of loss functions.

- **Representation learning and model pretraining.** There are quite a few recent works studying the factors that affect transferability of pretrained models, but the results are largely inconclusive. For example, the work Kornblith et al. (2019) argues that models pretrained on Imagenet with higher accuracy tend to perform better on other downstream tasks. However, such a conclusion has been challenged by more recent works (Kornblith et al., 2021; Nayman et al., 2022). These results showed that the training loss and diversity of the features could be more important factors of transferability than the pretrained accuracy. However, compared to our work, they only study few aspects (e.g., training loss) of deep network architectures that affect transferability, and they only focus on the diversity aspect on the source dataset. At the same time, the work Islam et al. (2021) showed that models learned using contrastive type of loss functions could have better transferability, and Wang & Isola (2020) showed that their representations are more uniform over hyperspheres. The architecture and depth of CNNs were also shown to impact transfer performance (Azizpour et al., 2015). Other work Zhang et al. (2019) exploited the importance of layers in overparameterized networks, suggesting the shallow and deep layers are more important in fine-tuning pretrained models for downstream tasks.

## 2 EVALUATING REPRESENTATIONS OF PRETRAINED MODELS VIA $\mathcal{NC}$

In this section, let us first give a brief overview of the $\mathcal{NC}$ phenomenon, upon which we introduce the metrics for evaluating the quality of learned representations for transfer learning in Section 3.

**Basics of deep neural networks.** Let us first introduce some basic notations by considering a multi-class (e.g., $K$ class) classification problem with finite training samples. Let $\{n_k\}_{k=1}^K$ be the number of training samples in each class and we assume the training samples are balanced $n = n_1 = n_2 = \cdots n_K$. Let $\boldsymbol{x}_{k,i}$ denote the $i$th input data in the $k$th class ($1 \le i \le n$, $1 \le k \le K$), and we use $\boldsymbol{y}_k \in \mathbb{R}^K$ to denote a one-hot training label with only the $k$th entry equal to unity. Thus, given any input data $\boldsymbol{x}_{k,i}$, deep network fits the corresponding (one-hot) training label $\boldsymbol{y}_k$ via

$$\boldsymbol{y}_k \approx \psi_{\boldsymbol{\Theta}}(\boldsymbol{x}_{k,i}) = \underbrace{\boldsymbol{W}_L}_{\text{linear classifier } \boldsymbol{W}} \cdot \underbrace{\phi_{\boldsymbol{\theta}}(\boldsymbol{x}_{k,i})}_{\text{feature } \boldsymbol{h}_{k,i} = \phi_{\boldsymbol{\theta}}(\boldsymbol{x}_{k,i})} + \boldsymbol{b}_L, \tag{1}$$

where $\boldsymbol{W} = \boldsymbol{W}_L$ is the last-layer linear classifier and $\boldsymbol{h}_{k,i} = \boldsymbol{h}(\boldsymbol{x}_{k,i}) = \phi_{\boldsymbol{\theta}}(\boldsymbol{x}_{k,i})$ denotes a deep hierarchical representation (or feature) of the input $\boldsymbol{x}_{k,i}$. Here, for a $L$-layer deep network $\psi_{\boldsymbol{\Theta}}(\boldsymbol{x})$, each layer is composed of an affine transformation, followed by a nonlinear activation $\sigma(\cdot)$ and normalization functions (e.g., BatchNorm (Ioffe & Szegedy, 2015)). We use $\boldsymbol{\Theta}$ to denote all network parameters of $\psi_{\boldsymbol{\Theta}}(\boldsymbol{x})$ and $\boldsymbol{\theta}$ to denote the network parameters of $\phi_{\boldsymbol{\theta}}(\boldsymbol{x})$. Additionally, we use

$$\boldsymbol{H} = [\boldsymbol{H}_1 \quad \boldsymbol{H}_2 \quad \cdots \quad \boldsymbol{H}_K] \in \mathbb{R}^{d \times N}, \quad \boldsymbol{H}_k = [\boldsymbol{h}_{k,1} \quad \cdots \quad \boldsymbol{h}_{k,n}] \in \mathbb{R}^{d \times n}, \ 1 \le k \le K,$$

to denote all the features in the matrix form. Additionally, we write the class mean for each class as

$$\overline{\boldsymbol{H}} := \begin{bmatrix} \overline{\boldsymbol{h}}_1 & \cdots & \overline{\boldsymbol{h}}_K \end{bmatrix} \in \mathbb{R}^{d \times K}, \quad \text{and} \quad \overline{\boldsymbol{h}}_k := \frac{1}{n} \sum_{i=1}^n \boldsymbol{h}_{k,i}, \quad 1 \le k \le K.$$

Accordingly, we denote the global mean of $\boldsymbol{H}$ as $\boldsymbol{h}_G = \frac{1}{K} \sum_{k=1}^N \overline{\boldsymbol{h}}_k$.

**A review of neural collapse.** Over the training dataset $\{\boldsymbol{x}_{k,i}, \boldsymbol{y}_k\}$, it has been widely observed that last-layer features $\boldsymbol{H}$ and classifiers $\boldsymbol{W}$ of a trained network exhibit simple but elegant mathematical structures (Papyan et al., 2020; Papyan, 2020), that we highlight two key properties below[2]

---

[2] Additionally, self-duality convergence has also been observed in the sense that $\boldsymbol{w}_k = c'\overline{\boldsymbol{h}}_k$ for some $c' > 0$, but this is not the main focus of this work.

- **Intra-class variability collapse:** for each class, the last-layer features collapse to their means,

$$\boldsymbol{h}_{k,i} \to \overline{\boldsymbol{h}}_k, \quad \forall\, 1 \le i \le n,\ 1 \le k \le K. \tag{2}$$

- **Maximum between-class separation:** the class-means $\left\{\overline{\boldsymbol{h}}_k\right\}_{k=1}^K$ centered at their global mean $\boldsymbol{h}_G$ are not only linearly separable, but are actually maximally distant and they form a Simplex Equiangular Tight Frame (ETF): for some $c > 0$, $\overline{\boldsymbol{H}} = \begin{bmatrix} \overline{\boldsymbol{h}}_1 - \boldsymbol{h}_G & \cdots & \overline{\boldsymbol{h}}_K - \boldsymbol{h}_G \end{bmatrix}$ satisfies

$$\overline{\boldsymbol{H}}^\top \overline{\boldsymbol{H}} \;=\; \frac{cK}{K-1}\left(\boldsymbol{I}_K - \frac{1}{K}\boldsymbol{1}_K\boldsymbol{1}_K^\top\right). \tag{3}$$

Recent work shows that $\mathcal{NC}$ persists across a range of canonical classification problems, on different loss functions (e.g., CE (Papyan et al., 2020; Fang et al., 2021), MSE (Mixon et al., 2020; Zhou et al., 2022; Han et al., 2022), SupCon (Fang et al., 2021; Graf et al., 2021)), on different neural network architectures (e.g., VGG (Simonyan & Zisserman, 2014), ResNet (He et al., 2016), and DenseNet (Huang et al., 2017)), and on a variety of standard datasets (e.g., MNIST (LeCun et al., 2010), CIFAR (Krizhevsky et al., 2009), and ImageNet (Deng et al., 2009)). As we observe from above, although the maximum between-class separation suggests the learned features are discriminative in (3), the intra-class variability collapse to a single dimension in (2) implies that the network is memorizing the labels rather than preserving the intrinsic structures of the data. As such, the loss of information of the input data could be detrimental for feature transferability.

**Measuring feature quality via $\mathcal{NC}$ metrics.** Based upon above discussion, the study of $\mathcal{NC}$ offers us new metrics to evaluate the transferability of pre-trained models.

- **Variability and separation collapses (Papyan et al., 2020; Zhu et al., 2021).** We can measure the variability and separation collapse via

$$\mathcal{NC}_1 \;:=\; \frac{1}{K}\operatorname{trace}\left(\boldsymbol{\Sigma}_W \boldsymbol{\Sigma}_B^\dagger\right), \tag{4}$$

by measuring the magnitude of the within-class covariance $\boldsymbol{\Sigma}_W \in \mathbb{R}^{d \times d}$ of the learned features compared to the inter-class covariance $\boldsymbol{\Sigma}_B \in \mathbb{R}^{d \times d}$, where

$$\boldsymbol{\Sigma}_W \;:=\; \frac{1}{nK}\sum_{k=1}^K \sum_{i=1}^n \left(\boldsymbol{h}_{k,i} - \overline{\boldsymbol{h}}_k\right)\left(\boldsymbol{h}_{k,i} - \overline{\boldsymbol{h}}_k\right)^\top, \quad \boldsymbol{\Sigma}_B \;:=\; \frac{1}{K}\sum_{k=1}^K \left(\overline{\boldsymbol{h}}_k - \boldsymbol{h}_G\right)\left(\overline{\boldsymbol{h}}_k - \boldsymbol{h}_G\right)^\top,$$

and $\boldsymbol{\Sigma}_B^\dagger$ denotes the pseudo inverse of $\boldsymbol{\Sigma}_B$. Here, $\boldsymbol{\Sigma}_B^\dagger$ serves as an normalization for $\boldsymbol{\Sigma}_W$ to capture the relativity between the two covariances. Intuitively, if the features are more collapse to their means, the smaller $\mathcal{NC}_1$ is; on the other hand, with the same $\boldsymbol{\Sigma}_W$, if the features have more separated class means, $\mathcal{NC}_1$ would also be smaller. Since the metric involves pseudo inverse of $\boldsymbol{\Sigma}_B$, computation of such a metric in (4) could be intractable when the feature dimension is large (for huge models).

## 3 METHODS & EXPERIMENTS

In the following, we will utilize the above metrics to experimentally evaluate the quality of learned representations in terms of transferability, to corroborate our claims in Section 1. More specifically, in Section 3.1 we focus on pre-training, that we demonstrate a positive correlation between $\mathcal{NC}$ metrics and transfer accuracy on pre-trained dataset. In Section 3.2, we turn our attention to downstream tasks, where we discover the $\mathcal{NC}$ metrics and the associated transfer accuracy are negatively correlated. Finally, in Section 3.3, based on the above findings, we propose a simple and efficient fine-tuning method in a principled manner.

**Experimental setups.** We pre-train ResNet18 and ResNet50 (He et al., 2016) models on Cifar-100 (Krizhevsky et al., 2009) and MiniImageNet (Vinyals et al., 2016) datasets. Unless otherwise specified, we pre-train the models for 200 epochs using the SGDR learning rate scheduler (Loshchilov & Hutter, 2017) with initial learning rate 0.1 and minimum learning rate 0.0001. More details are postponed to Appendix A.2.

### 3.1 STUDY OF $\mathcal{NC}$ & TRANSFER ACCURACY ON MODEL PRETRAINING

We begin our investigation by studying the relationship between $\mathcal{NC}$ metrics on *pre-training dataset* and transfer accuracy. To certain extent, we find the two are positively correlated: larger $\mathcal{NC}$ metrics often leads to better transfer accuracy. Our key intuition is that if the learned representations are less collapse on the pre-trained data, they better preserve the intrinsic structures of the input data.

| Training | MSE (w/o proj.) | Cross-entropy (w/o proj.) | SupCon (w/ linear proj.) | SupCon (w/ mlp proj.) |
|---|---|---|---|---|
| $\mathcal{NC}_1$ (Cifar-100) | 0.001 | 0.771 | 0.792 | 2.991 |
| Transfer Acc. | 53.96 | 71.2 | 69.89 | 79.51 |

Table 1: **Transfer learning results & model $\mathcal{NC}_1$ comparison among different training settings.** ResNet18 models are pre-trained on the Cifar-100 dataset and transfered on Cifar-10. We use proj. to denote projection head.

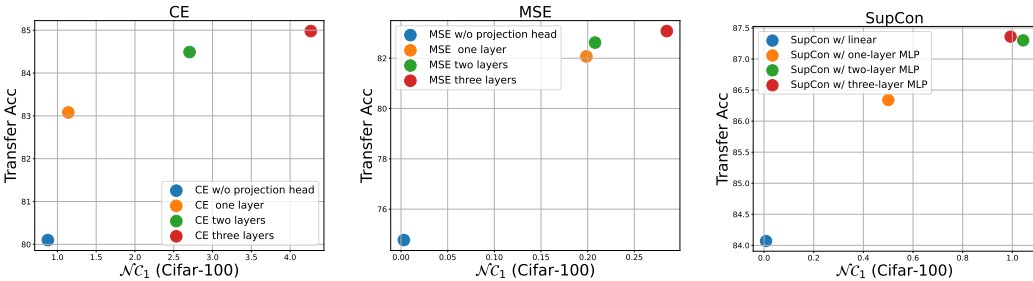

Figure 1: **Trend of $\mathcal{NC}_1$ during training and transfer learning accuracy of the pretrained models.** ResNet50 models are pretrained using Cifar-100 dataset with CE loss (Left), MSE loss (Middle) and SupCon loss (Right). Models are pretrained with different numbers of layers for projection heads and transfered on the Cifar-10 dataset.

**Choices of training losses and architecture impact feature variability and hence transferability.** First, we show that the choice of training losses and design of architecture substantially affects the collapse of the features on the penultimate layer, and hence the transfer accuracy.[3] To show this, we pre-train ResNet18 models on the Cifar-100 dataset with three different choices of loss functions (CE, MSE (Hui & Belkin, 2020), and supervised contrastive loss (SupCon) (Khosla et al., 2020)), and then test the transfer accuracy on the Cifar-10 dataset. For the SupCon (Khosla et al., 2020), we follow the original setup that uses a nonlinear multi-layer perception (MLP) module as a projection head after the ResNet18 encoder. Once the model has been pre-trained, the projection head will be abandoned and only the encoder network will be utilized as the model for downstream tasks. In Table 1, we summarize the results of model $\mathcal{NC}_1$ and transfer accuracy for different training scenarios. As we can see, the model trained with the MSE loss is severely collapsed on the source dataset and has the worst transfer accuracy. On the other hand, the model trained using the SupCon with an MLP projection head is less collapse compared to the other models and has superior transfer performance. This observation is consistent with (Islam et al., 2021) that SupCon leads to better transferability than CE. Here, we further show that this is because SupCon learns relatively more diverse features (larger $\mathcal{NC}_1$) than CE. However, given both SupCon and CE learn the same $\mathcal{NC}$ representations in the so-called unconstrained features model (Fang et al., 2021; Zhu et al., 2021; Graf et al., 2021), it raises an interesting question on why the features learned by SupCon are more diverse. We further address this question in the following.

**Projection layers in pre-training increase feature diversity for better transferability.** We conjecture that the inclusion of an MLP projection head serves as an important role for the better performance of SupCon. This is based on the recent works (Papyan, 2020; Hui et al., 2022; He & Su, 2022) that show the within-class variability collapse actually happens progressively from shallow to deep layers: the closer to the final layer, the severer the variability collapse (i.e., the smaller the $\mathcal{NC}_1$ of that layer). Therefore, *the additional MLP projection layers prevent severe variability collapse of the encoder network*. We empirically verify this conjecture by replacing the MLP by a linear layer for the projection head in SupCon and then train the model under the same setting. We observe from Table 1 that SupCon with a linear projection head indeed achieves similar $\mathcal{NC}$ and transfer accuracy as CE. Note that the layer-wise progressive collapse is universal across the choice of training losses, the usage of projection head is not limited to contrastive losses. Thus, we can also improve the transferability for supervised pre-training models with the CE / MSE loss by adding a MLP projection head. To verify this, we pre-train ResNet-50 models (we use it for better performance) on the Cifar-100 dataset. We report the $\mathcal{NC}$ and transfer accuracy for different choices of projection heads. In

---

[3]For the choices of loss, the works (Islam et al., 2021; Kornblith et al., 2021) have similar observations, where different choices of training losses for pre-training lead to different transfer performance.

Figure 1, we can observe that the inclusion of projection heads substantially increase the diversity of the representations and the transfer accuracy compared with that of no projection head. We note that the usage of projection head is first introduced and popularized for self-supervised learning (Chen et al., 2020; Chen & He, 2021). But the reasons behind the success of such a method remain as mystery. Here, we demystify the underlying mechanism of projection head based upon progressive variability collapse, where our experiments show that the usage of projection heads is not limited to pre-training by contrastive losses.

**Pre-training $\mathcal{NC}$ does not always explain transfer accuracy.** So far, we have shown that models that are very collapsed on the source dataset have inferior transferability, which is aligned with the observation in (Hui et al., 2022). However, does this imply the truth of the opposite direction: models that are not collapsed on the source dataset will always transfer better? The answer is obviously no. A naive counter example would be an untrained model with randomly initialized weights, which will not collapse but also has poor transfer accuracy. Because the reason for the increase in $\mathcal{NC}_1$ could be either from the increase of feature diversity within each class or from the loss of discriminate power between classes. To more comprehensively characterize this relationship, we pre-train ResNet50 models on the Cifar-100 dataset using different levels of data augmentations and different adversarial training (Madry et al., 2018; Salman et al., 2020; Deng et al., 2021) strength, and then test the transfer accuracy on the Cifar-10 dataset, where the result is summarized in Figure 2. We observe that positive relationship between the level of collapse on the source dataset and the transferability only holds up to certain threshold[4] based upon the $\mathcal{NC}_1$ metric. As shown in Figure 2, if the feature variability is beyond the threshold, the transfer accuracy decreases as the feature variability increases (i.e., $\mathcal{NC}_1$ increases).

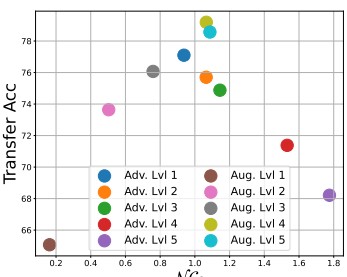

Figure 2: $\mathcal{NC}_1$ **vs. Transfer learning accuracy.** Models are pretrained using the Cifar-100 dataset with different data augmentation levels and adversarial training strength, transfer accuracy is evaluated on the Cifar-10 dataset.

### 3.2 Study of $\mathcal{NC}$ & Transfer Accuracy on Downstream Tasks

Second, given the pre-trained models, on *downstream data* we experimentally investigate the relationship between $\mathcal{NC}$ metrics of their representations and transfer accuracy. Transferring pre-trained large models to smaller downstream tasks has become the dominant approach in both vision (Dosovitskiy et al., 2021) and language (Devlin et al., 2019) domains. Also, in many cases the source data for pre-training is unavailable,[5] we can only evaluate the representation quality based upon the downstream data.

Here, to control the factors for affecting our study, the experimental setting is quite simple: for each downstream task, we *freeze* the whole pre-trained model with no fine-tuning, and for the model we only train the linear classifier on the downstream data. On the contrary to model pre-training, our discovery is that the transfer accuracy is negatively correlated with the $\mathcal{NC}$ metrics on downstream data. In other words, the more collapsed the representations of the layer on the downstream data or simpler dataset, the better the transfer accuracy. Moreover, as we show in the following, this phenomenon is quite universal: it not only happens across pre-trained models with different pre-training strategies, but also across different layers of the same pre-trained model.

**Pre-trained models with more collapsed last-layer features result in better transferability.** To validate our statement, we pre-train different ResNet50 models on the Cifar-100 dataset by using different levels of data augmentations and different levels of adversarial training strengh. Once a model is pre-trained, we test its transfer accuracy on 4 downstream datasets: Cifar-10 (Krizhevsky et al., 2009), FGVS-Aircraft (Maji et al., 2013), DTD (Cimpoi et al., 2014) and Oxford-IIIT-Pet (Parkhi et al., 2012) datasets. In Figure 3, we observe that the $\mathcal{NC}_1$ on Cifar-10 dataset has a negative (almost linear) correlation with the transfer accuracy on different downstream tasks. The more collapse of feature on the Cifar-10 dataset (i.e, the smaller the $\mathcal{NC}_1$ metric is), the higher the

---

[4]We note the work (Kornblith et al., 2021) studied similar correlation by introducing a notion called class separation. However, the work has not studied the aspects when the class separation approaches 0 which is similar to entirely not collapsing with the $\mathcal{NC}$ notions.

[5]e.g., JFT dataset (Sun et al., 2017), which is used in the pretraining of Vision Transformer, is not publicly available.

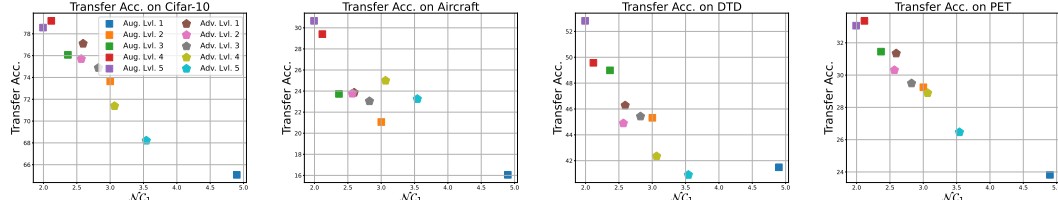

Figure 3: **Transfer accuracy on different downstream tasks and** $\mathcal{NC}_1$**.** We pre-train ResNet50 models on Cifar-100 using different levels of data augmentation or adversarial training. $\mathcal{NC}_1$ is measured on the downstream Cifar-10 dataset.

transfer accuracy.[6] As such, we observe that the $\mathcal{NC}_1$ metric on Cifar-10 evaluated on pre-trained models can serve as a good performance indicator for the transfer accuracy on downstream tasks.

The opposite relationships between $\mathcal{NC}$ metrics and transfer accuracy on pre-trained data and on downstream data may seem contradictory at first glance. However, for pre-training, as we explained in Section 3.1, we desire less collapsed models on source data so that the learned features can capture the structure of the input data. On the contrary, on downstream data, the $\mathcal{NC}$ metrics act as a measure on model's ability of representing the input data to fit the output label. As such, the less collapse on pre-trained data, the better representation power of the pre-trained model. Therefore, the larger $\mathcal{NC}_1$ on pre-trained data translates to smaller $\mathcal{NC}_1$ on downstream data and better transfer accuracy.

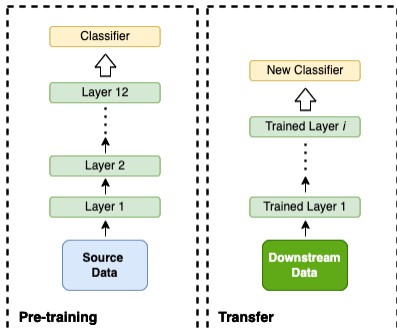

Figure 4: **Illustration of layer-wise transfer learning.** We pre-train a MLP model and use the outputs from each layer to do transfer learning.

**Layers with more collapsed output features result in better transferability.** More intriguingly, the phenomenon we observed above not only happens across different pre-trained models, but also happens across (the outputs of) different layers on exactly the same pre-trained model. More precisely, given the same pre-trained model, as shown in Figure 4, we use the output of each individual layer as a "feature extractor", and we test the transfer accuracy of the given layer by training a linear classifier on top of it. Surprisingly, if the outputs of the layer are more collapsed, using the corresponding features leads to better transfer accuracy, which happens regardless of the layer's depth.

To corroborate our claim, we pre-train a 12-layer MLP network with the same output dimension for every layer,[7] and we evaluate the $\mathcal{NC}_1$ metric on the layer's output feature upon the downstream data. To evaluate the transfer accuracy of the given layer, we train a linear classifier on top of the layer's output based upon downstream training data. As we observe in Figure 5, the smaller $\mathcal{NC}_1$ of the features of a given layer, the better transfer accuracy we get. We can observe a near linear relationship between $\mathcal{NC}_1$ and transfer accuracy from Figure 5 (Right). Thus, the transfer accuracy is more correlated with the variability collapse upon the layer rather than the depth of the layer. Furthermore, the phenomenon we illustrated above holds more universally beyond the vanilla MLP architecture. To demonstrate this, we conducted the same experiment on ViT-B (vision transformer base model) (Dosovitskiy et al., 2021) by using a pre-trained checkpoint released online [8]. Similarly, we compute the $\mathcal{NC}_1$ of each layer and train a linear classifier for each of the encoder layers in the ViT-B. The results are reported in Figure 6, where we can observe the same trend and more linear relationship compared to the MLP experiment.

---

[6]When evaluating the relationship between $\mathcal{NC}_1$ and transfer accuracy measured on the same downstream dataset, the correlation is not as strong as we find on Cifar-10 dataset, we leave the results and discussion in Appendix A.3.

[7]We choose the MLP architecture instead of other architectures (e.g., ResNet) is to get rid of the influence of feature dimensions for fair comparisons across layers. For MLP, we can freely choose the the output dimension of each layer. However, the layer's output dimension in ResNet is usually fixed and varies significantly across different depth.

[8]The Vit-B model checkpoint we used could be found here.

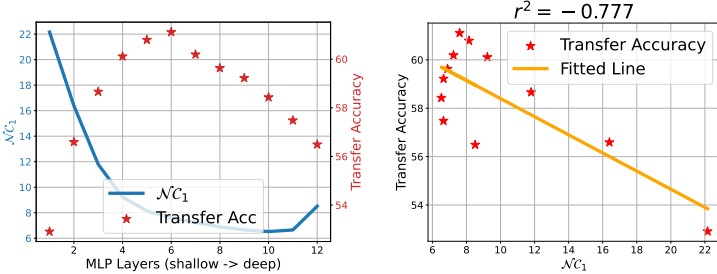

Figure 5: $\mathcal{NC}_1$ **and transfer learning accuracy of different layers from a pretrained MLP model (Left); Nearly linear relationship between transfer learning accuracy and** $\mathcal{NC}_1$ **(Right).** The 12 layer, 3072 hidden dimension MLP model is pretrained on the Cifar-100 dataset and transfered on the Cifar-10 dataset. $\mathcal{NC}_1$ is evaluated on the Cifar-10 dataset.

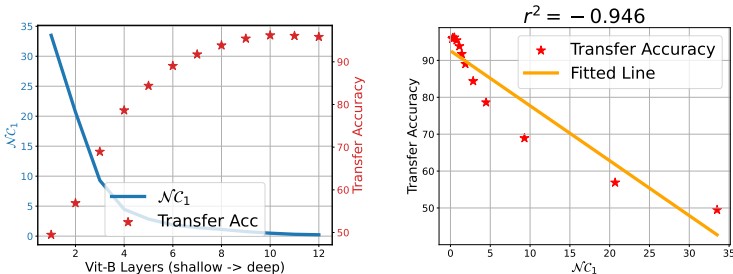

Figure 6: $\mathcal{NC}_1$ **and transfer learning accuracy of different layers from a pretrained ViT-B model (Left); Nearly linear relationship between transfer learning accuracy and** $\mathcal{NC}_1$ **(Right).** We use a pretrained ViT-B model and transfered on the Cifar-10 dataset. $\mathcal{NC}_1$ is evaluated on the Cifar-10 dataset.

## 3.3 A SIMPLE & EFFICIENT FINE-TUNING STRATEGY FOR IMPROVING TRANSFERABILITY

Finally, we show that the phenomenon we discovered in Section 3.2 can be very useful for designing simple yet more efficient fine-tuning strategies without sacrificing the performance. For vision tasks, model transfer learning (Kornblith et al., 2019; 2021; Deng et al., 2021; Salman et al., 2020) typically adopts two strategies: (1) **fixed feature training**, use pre-trained model up to the penultimate layer as a feature extractor and only train a new linear classifier on top of the features for a downstream task; (2) **full model fine-tuning**, use the pre-trained model as an initialization and fine-tune the whole model to fit a downstream task. However, the full fine-tuning could be very expensive for large models, while fixed feature training often results in worse performance without adapting the model to the downstream data. Thus, the middle case, fine-tuning only a selected subset of layers, could be a promising approach for balancing the computation and performance, while it is seldom studied for vision problems (Utrera et al., 2021; Shen et al., 2021). Based upon the correlation between penultimate layer collapse and the transfer learning performances we discovered in Section 3.2, our conjecture is that

*The topmost transfer accuracy can be achieved by selectively fine-tuning the layers such that the features of penultimate layer are the most collapsed on the downstream training data.*

Towards this goal, in this work we propose a simple strategy to increase the level of collapse of the penultimate layer – fine-tuning one additional layer besides the final linear classifier. We notice that by fine-tuning only one additional layer, the penultimate $\mathcal{NC}$ can be substantially decreased along with better transfer performance. More specifically, in table 2, we examine this simple strategy on various architectures and source / downstream datasets, where for each architecture, we pick several layers from the network, fine-tune each of them with the linear classifier and report the maximum transfer accuracy. As we observe, the additional one layer fine-tuning leads to substantial performance gain compared with linear probing regardless of pre-training scenarios and downstream datasets.

| Backbone | ResNet18 | | ResNet50 | | Vit-B | | | |
|---|---|---|---|---|---|---|---|---|
| **Dataset** | Cifar-10 | Cifar-100 | Cifar-10 | Cifar-100 | Aircraft | DTD | PET | |
| **Linear Probe** | 43.12 | 23.84 | 49.70 | 29.71 | 43.65 | 73.88 | 92.23 | |
| **Layer FT** | 82.00 | 54.03 | 85.03 | 59.39 | **65.83** | 77.13 | 93.02 | |
| **SCL FT** | 84.83 | **58.51** | 86.14 | **64.90** | 65.80 | **77.34** | **93.19** | |
| **Full Model FT** | **85.61** | 55.75 | **88.54** | 53.42 | 64.57 | 76.49 | 93.02 | |

Table 2: **Transfer learning results for Linear probing, layer fine-tuning, SCL fine-tuning and full model fine-tuning on various downstream datasets.** We use released ResNet models pretrained on ImageNet-1k (Deng et al., 2009) and ViT-B model pre-trained on JFT (Sun et al., 2017) and ImageNet-21k (Ridnik et al., 2021) datasets.

Despite the success of the above simple approach, we note that fine-tuning a shallow layer could only implicitly impact the $\mathcal{NC}$ dynamics of the penultimate layer because all the intermediate layers remain un-trained. To make the impact more direct, as shown in Figure 7, we add a skip connection from the fine-tuned layer features to the penultimate layer features and use the combined features to serve as the inputs for the final linear classifier. Such method enables the network to more effectively fine-tune the selected layer by explicitly passing the learned information to the classifier without suffering the information loss through the cascade of intermediate layers. On the other hand, since we are using a skip connection rather than discarding the top layers and using the fine-tuned layer features directly for classification, the network still benefits from being deep. We term our method SCL (**S**kip **C**onnection **L**ayer) fine tuning and compare its results with layer fine-tuning, linear probe and full model fine-tuning in Table 2. We observe that SCL fine-tuning nearly always outperforms layer fine-tuning and achieves comparable or even better results compared with full model fine-tuning.

## 4 DISCUSSION & CONCLUSION

**Twofold relationship between $\mathcal{NC}$ and transferability.** In this work, we examine the relationship between $\mathcal{NC}$ and transferability through various aspects. Previous work (Hui et al., 2022) points out that $\mathcal{NC}$ is mainly an optimization phenomenon which does not necessarily relate to generalization (or transferability). Our work, on one hand, corroborates with the finding that pretraining $\mathcal{NC}$ does not always suggest better transferbality, but also shows a positive correlation between pretraining $\mathcal{NC}$ and transferability to certain extent. On the other hand, our work also shows that downstream $\mathcal{NC}$ on a dataset where $\mathcal{NC}$ is well-defined correlates with the transfer performances across different datasets and thus could be a general indicator for the transferability. This suggests that $\mathcal{NC}$ may not be merely an optimization phenomenon. An important future direction we will pursue is to theoretically understand the connection between transferability and $\mathcal{NC}$.

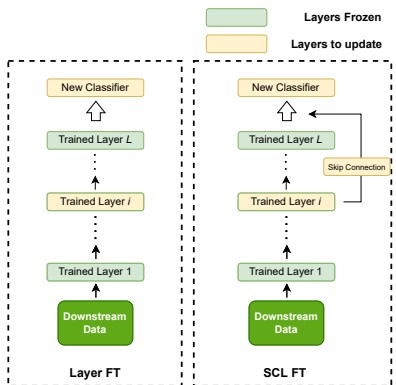

Figure 7: **Illustration of SCL fine-tuning.**

**Boost model transferability by insights from $\mathcal{NC}$.** Our findings can be used to improve model transferability through the following two perspective. First, the positive correlation between pretraining $\mathcal{NC}$ and transferability suggests that increase diversity of features to certain extent can improve transferability. This can be achieved by popular techniques such as multi-layer projection heads and data augmentation. We believe other principled approaches could also be developed by explicitly working on the geometry of the representations. Second, by demonstrating the close correlation between downstream $\mathcal{NC}$ and associated transfer accuracy, we are able to make simple yet effective strategies to do model transfer learning. However, our simple approach is by no means the optimal method to exploit such relationship. We believe there are more powerful approaches that could utilize this phenomenon better and thus achieves better transferability. We leave this as a future work.

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

# A APPENDIX

## A.1 OTHER METRICS FOR MEASURING $\mathcal{NC}$

**Class-distance normalized variance (CDNV) Galanti et al. (2022).** To alleviate the computational issue of $\mathcal{NC}_1$, the *class-distance normalized variance* (CDNV) introduced in (Galanti et al., 2022) provides an alternative metric that is inexpensive to evaluate. Let $\mathcal{X}$ denotes the space of the input data $x$ and let $\boldsymbol{Q}_k$ be the distribution over $\mathcal{X}$ conditioned on the class $k$. For two different classes with $\boldsymbol{Q}_i$ and $\boldsymbol{Q}_j$ ($i \neq j$), the CDNV metric can be described by the following equation: $V_{\phi_\theta}(\boldsymbol{Q}_i, \boldsymbol{Q}_j) = \frac{\mathrm{Var}_{\phi_\theta}(\boldsymbol{Q}_i) + \mathrm{Var}_{\phi_\theta}(\boldsymbol{Q}_j)}{2||\mu_{\phi_\theta}(\boldsymbol{Q}_i) - \mu_{\phi_\theta}(\boldsymbol{Q}_j)||_2^2}$, where $\mu_{\phi_\theta}(\boldsymbol{Q}_k) = \mathbb{E}_{x \sim \boldsymbol{Q}_k}[\phi_\theta(x)]$ denotes the class-conditional feature mean and $\mathrm{Var}_{\phi_\theta}(\boldsymbol{Q}_k) = \mathbb{E}_{x \sim \boldsymbol{Q}_k}[||\phi_\theta(x) - \mu_{\phi_\theta}(\boldsymbol{Q}_k)||^2]$ denotes the feature variance for the distribution $\boldsymbol{Q}_k$. Although the exact expectation is impossible to evaluate, we can approximate them via their empirical means and empirical variances on the given training samples, so that

$$\widehat{V}_{\phi_\theta}(\boldsymbol{Q}_i, \boldsymbol{Q}_j) = \frac{\widehat{\mathrm{Var}}_{\phi_\theta}(\boldsymbol{Q}_i) + \widehat{\mathrm{Var}}_{\phi_\theta}(\boldsymbol{Q}_j)}{2||\widehat{\mu}_{\phi_\theta}(\boldsymbol{Q}_i) - \widehat{\mu}_{\phi_\theta}(\boldsymbol{Q}_j)||^2}, \tag{5}$$

$$\widehat{\mu}_{\phi_\theta}(\boldsymbol{Q}_k) = \frac{1}{n_k}\sum_{i=1}^{n_k}\phi_\theta(x_{k,i}), \quad \widehat{\mathrm{Var}}_{\phi_\theta}(\boldsymbol{Q}_k) = \frac{1}{n_k}\sum_{i=1}^{n_k}||\phi_\theta(x_{k,i}) - \mu_{\phi_\theta}(\boldsymbol{X}_k)||^2 \tag{6}$$

To characterize the overall degree of collapse for a model, we can use the average CDNV between all pairwise classes (i.e., $\mathrm{Avg}_{i \neq j}[\widehat{V}_{\phi_\theta}(\boldsymbol{Q}_i, \boldsymbol{Q}_j)]$). If a model achieves perfect $\mathcal{NC}$, obviously we have $\mathrm{Avg}_{i \neq j}[\widehat{V}_{\phi_\theta}(\boldsymbol{Q}_i, \boldsymbol{Q}_j)] = 0$. Because the CDNV metric is purely norm-based, computation complexity scales linearly with the feature dimension $d$, so that it serves as a good surrogate for $\mathcal{NC}_1$ when the feature dimension $d$ is large.

**Numerical rank of the features $H$.** The $\mathcal{NC}_1$ does not directly reveal the dimensionality of the features spanned for each class. Measuring rank of the features ($\boldsymbol{H}_k$) is more suitable. However, the calculations for both $\mathcal{NC}_1$ and rank are expensive when feature dimension gets too large. Thus, we introduce *numerical rank* (Timor et al., 2022) as an approximation

$$\widetilde{\mathrm{rank}}(\boldsymbol{H}) := \frac{1}{K}\sum_{k=1}^{K}\|\boldsymbol{H}_k\|_F^2 / \|\boldsymbol{H}_k\|_2^2,$$

where $\|\cdot\|_F$ represents the Frobenius norm and $\|\cdot\|_2$ represents the Spectral norm. $\widetilde{\mathrm{rank}}(\boldsymbol{H})$(*numerical rank*) could be seen as an estimation of the true rank for any matrix. Note that for calculating the Spectral norm, we use the Power Method to find an approximation. The metric is evaluated by averaging over all the classes. It is expected that the smaller $\widetilde{\mathrm{rank}}(\boldsymbol{H})$ is, the more collapsed the features are to their class means.

## A.2 TECHNICAL DETAILS FOR SECTION 3.

In the main paper, for the ease of presentation, some technical details in the experiments are not presented explicitly. We discuss the missing implementation details here.

**Experimental set-up for Section 3.1** In Figure 1, we pretrain ResNet50 models with different number of projection layers using Cifar-100 and MiniImageNet datasets for 200 epochs. Then we use the learned model to do transfer learning on Cifar-10. The $\mathcal{NC}_1$ and CDNV are then evaluated on the source dataset.

**Experimental set-up for Section 3.2** In Figure 2, we pretrain ResNet50 models using different levels of data augmentation and adversarial training. For data augmentation, we consider Random-Crop, RandomHorizontalFlip, ColorJitter and RandomGrayScale. We add an additional augmentation for each level. I.e., for augmentation level 1, we don't add any data augmentation, for level 2, we add RandomCrop, for level 3, we add RandomCrop + RandomHorizontalFlip, and etc. For adversarial training, we follow the $\ell_\infty$ norm bounded adversarial training framework in (Madry et al., 2018) with 5 levels of attack size: $\left\{\frac{1}{255}, \frac{2}{255}, \frac{3}{255}, \frac{5}{255}, \frac{8}{255}\right\}$.

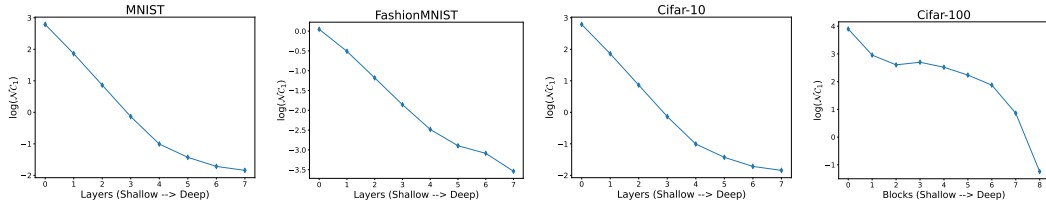

Figure 9: **Change of $\mathcal{NC}_1$ dynamic for trained MLP / ResNet18 models.** The MLP model is trained on MNIST (LeCun et al., 2010), FashionMNIST (Xiao et al., 2017), Cifar-10, respectively and the ResNet18 model is trained on Cifar-100.

With the same pretraining setup as in Figure 2, we transfer the learned models on 4 different downstream datasets: Cifar-10, FGVS-Aiscraft, DTD and Oxford-IIIT-Pet. We note that there are many benchmark datasets that we can potentilly use, we choose these 4 datasets because the number of samples for each class is balanced in these datasets, which is the same scenario where $\mathcal{NC}$ is first studied (Papyan et al., 2020).

In Figure 6, we use the Vit-B32 model with pretrained weights released online. For each encoder layer in Vit-B32, the outputs are of size 145 (# of patches + an additional classification token) × 768 (hidden dimension). For the layer-wise transfer learning experiment, we first do a average pooling on the 145 patches and then train a linear classifier with input dimension 768 on top of each encoder layer.

**Experimental set-up for Section 3.3** For Table 2, we use a wide variety of experimental setups, including different model architectures, pretraining datasets and downstream datasets. We then compare the performance between linear probing, layer fine-tuning, SCL fine-tuning and full model fine-tuning. For ResNet models, we consider each block as a fine-tuning unit and fine-tune the first block of each layer (e.g., as shown in Figure 8, ResNet18 has 4 layers where each layer has 2 blocks, we then fine-tune on the first block of each layer). For Vit-B32 model,

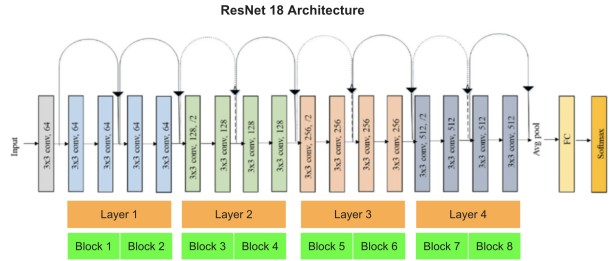

Figure 8: **Fine-tuning unit of ResNet.** Image of ResNet18 from (Ramzan et al., 2019)

we treat each of the 12 encoder layer as a fine-tuning unit. In terms of skip connection, for ViT-B32 model, since the feature dimension from each layer remain constant, the skip connection could be directly applied for the features of the fine-tuned layer and the penultimate layer. However, for ResNet models, the number of channels and the feature dimension change across layers. Therefore, to calculate the skip connection, we first do an adaptive average pooling on the fine-tuned layer features to make each channel has only one entry; then we further do a 1 × 1 convolution to make the number of channels match with the penultimate layer features. Finally, we do a BatchNorm (Ioffe & Szegedy, 2015) on the processed fine-tuned layer features and apply the skip connection.

## A.3 ADDITIONAL RESULTS

### A.3.1 ADDITIONAL RESULTS FOR SECTION 3.1

In Section 3.1, we conjecture that the success of adding projection layers on transfer learning can be attributed to the progressively within-class variability collapse. Recently, a work (He & Su, 2022) further found that $\log(\mathcal{NC}_1)$ decays linearly across layers. We train MLP and ResNet18 models on different datasets and validate their observation in Figure 9. Such phenomenon accompanies our results in Section 3.1 and explains why adding projection heads is useful in the perspective of feature collapse.

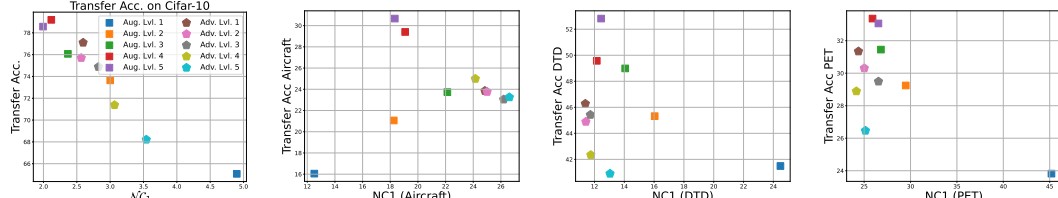

Figure 10: **Transfer accuracy on different downstream tasks and** $\mathcal{NC}_1$**.** Transfer accuracy and $\mathcal{NC}_1$ are measured on the same downstream datasets.

| FT method | Linear Probe | Block 1 | | Block 3 | | Block 5 | | Block 7 | | Full Model |
|---|---|---|---|---|---|---|---|---|---|---|
| | | Layer FT | SCL FT | Layer FT | SCL FT | Layer FT | SCL FT | Layer FT | SCL FT | |
| **Cifar-10** | 43.12 | 65.03 | 82.33 | 80.84 | 84.83 | 82.00 | 83.10 | 71.66 | 73.08 | **85.61** |
| **Cifar-100** | 23.84 | 35.98 | 53.52 | 46.26 | **58.51** | 54.03 | 57.72 | 45.85 | 46.97 | 55.75 |

Table 3: **Transfer learning performance of ResNet18 (pretrained on Imagenet) on downstream datasets with different fine tuning methods.**

### A.3.2 ADDITIONAL RESULTS FOR SECTION 3.2

In Section 3.2, we show that the $\mathcal{NC}_1$ on Cifar-10 dataset negative correlates with the transfer accuracy on different downstream tasks. Nevertheloss, in Figure 10, when we evaluate the relationship between the transfer accuracy and the associated $\mathcal{NC}_1$ on the same downstream dataset, we cannot find a strong correlation. We conjecture that the reason behind such mismatch is the similar with our analysis in Section 3.1: when the values of $\mathcal{NC}_1$ get too large, the metric starts to become less meaningful since the increase may come from either an expansion in the feature variance within each class, or a loss of discriminative power between classes.

### A.3.3 ADDITIONAL RESULTS FOR SECTION 3.3

In Table 2, for the clarity of presentation, we report the maximum transfer performance gained fine-tuning different layers. Here, we report the results for all of the fine-tuned layers in Table 3, Table 4 and Table 5 for ResNet18, ResNet50 and Vit-B32 respectively. We can observe that fine-tune middle layers nearly always give the best transfer performance.

Moreover, to confirm that SCL fine-tuning yields more collapsed penultimate layer features, we fine-tune a ResNet18 model trained with Cifar-100 dataset on Cifar-10 using the layer fine-tuning and SCL fine-tuning methods. We then visualize the change in the $\mathcal{NC}_1$ dynamic across network layers before and after fine-tuning in Figure 11 (Left). Fine-tuning a block would make the features from the block and all following blocks become more collapse on the downstream task, and adding skip connections make the collapse more severe and thus lead to more collapsed penultimate layers and potentially better transfer performance. In Figure 11 (Right), we plot the correlation between $\mathcal{NC}_1$ and the transfer accuracy. We notice the negative correlation is well-aligned with our observation in Figure 3. However, we note that although more collapsed penultimate features almost surely leads to better transferability for fixed feature transfer learning (linear probing), it is not always reliable in the case of model fine-tuning. This is because the more we change model parameters before the linear classifier, the model becomes more prune to remember the data-label relationship in the downstream datasets and hence make the fine-tuned model likely to overfit. Eventually when the full model is being fine-tuned, we would get neural collapsed model on the downstream training data as well but the collapse in this case may not translate to better performances on the test set. Such phenomenon happens especially severe for the datasets with limited number size, as shown in (Hui et al., 2022).

| FT method | Linear Probe | Block 1 | | Block 4 | | Block 8 | | Block 14 | | Full Model |
|---|---|---|---|---|---|---|---|---|---|---|
| | | Layer FT | SCL FT | Layer FT | SCL FT | Layer FT | SCL FT | Layer FT | SCL FT | |
| **Cifar-10** | 49.70 | 38.65 | 78.58 | 49.76 | 79.24 | 85.03 | 86.14 | 75.78 | 76.02 | **88.54** |
| **Cifar-100** | 29.71 | 6.98 | 55.94 | 25.02 | 59.36 | 59.39 | **64.90** | 52.29 | 53.56 | 53.42 |

Table 4: **Transfer learning performance of ResNet50 (pretrained on Imagenet) on downstream datasets with different fine tuning methods.**

| FT method | Linear Probe | Layer 1 | | Layer 4 | | Layer 7 | | Layer 10 | | Full Model |
|---|---|---|---|---|---|---|---|---|---|---|
| | | Layer FT | SCL FT | Layer FT | SCL FT | Layer FT | SCL FT | Layer FT | SCL FT | |
| **DTD** | 73.88 | 76.54 | 77.02 | 75.85 | 77.18 | 77.13 | **77.34** | 76.12 | 76.54 | 76.49 |
| **PET** | 92.23 | 92.42 | 92.23 | 92.67 | **93.19** | 92.94 | 93.13 | 93.02 | 93.13 | 93.01 |
| **Aircraft** | 43.65 | 57.64 | 56.50 | 64.93 | 62.35 | **65.83** | 65.80 | 62.80 | 62.32 | 64.57 |

Table 5: **Transfer learning performance of Vit-B32 (pretrained on JFT and ImageNet-21k) on downstream datasets with different fine tuning methods.**

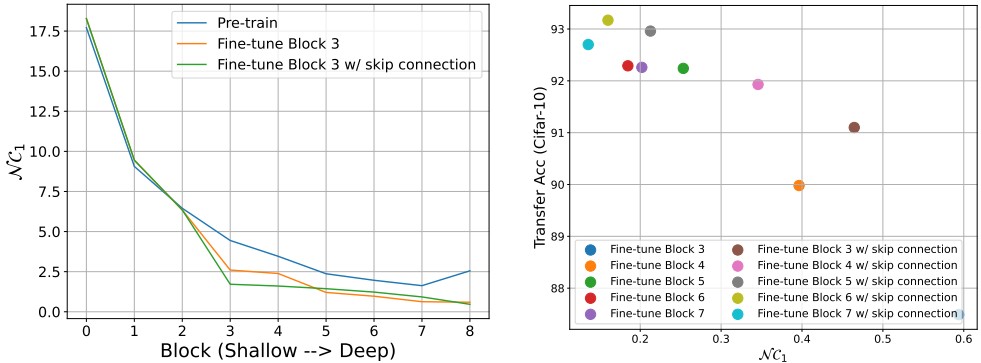

Figure 11: **Change of $\mathcal{NC}_1$ dynamic before and after fine-tuning (Left), Correlation between downstream $\mathcal{NC}_1$ and transfer accuracy (Right).** Transfer accuracy and $\mathcal{NC}_1$ are measured on the same downstream datasets.

