# OpenReview forum: "What Deep Representations Should We Learn? -- A Neural Collapse Perspective"
_ICLR.cc/2023/Conference — Submitted to ICLR 2023_

### Official Review · Reviewer_24Wm · 2022-10-23

**Confidence:** 2
**Correctness:** 2
**Technical Novelty And Significance:** 1
**Empirical Novelty And Significance:** 2
**Recommendation:** 3

**Clarity, Quality, Novelty And Reproducibility:**

This paper covers a very timely topic: Neural Collapse in relation to transfer learning. But the paper itself is not novel. The first conclusion is that higher diversity in a pretrained model leads to better transfer learning. I believe that this conclusion has already been shown in other works, including Kornblith et al. (2021); Nayman et al. (2022). The second conclusion is that more collapsed test-data features lead to a higher accuracy, which is a direct consequence of the NC phenomena.

In terms of quality, my main concern is that the experiments were not rigorously designed.


**Strength And Weaknesses:**

Strenth:

- The paper addressed a very important problem: How transfer learning can be performed in a principlined way, potentially through the lens of neural collapse.
- The authors provided a review that covers many relevant prior studies.
- The experiments contain some interesting (perhaps not very surprising) results.

Weaknesses:

- The paper was based on an interesting idea, but the execution is not convincing. Because the experiment setup is not very well described, I  can only fill in some details using my imagination. Overall, I have quite a few questions about the soundness of the experiment design.

In particular, the definition of the $\mathcal{NC}_1$ metric (Eq 4)  is not sufficiently motivated. To me, it seems more useful to consider the impact of two factors, $\Sigma_B$ and $\Sigma_W$ separately. The original NC paper already defined some quantitative metrics for convergence to simplex ETF; Why bother reinventing the wheels?

- The experiment results are mostly unsurprising, but the author's conclusions were sometimes not justified and may be misleading.

In particular, the discussion surrounding MLP projection layers does not look correct to me. NC is about the last-layer features in a terminal phase of training. The authors may have misunderstood the original NC paper.



**Summary Of The Paper:**

This paper empirically explored how the Neural Collapse (NC) phenomena may affect transfer learning. The authors proposed two metrics $\mathcal{NC}_1$ and CDNV to measure the degree of intra-class variability in relation to between-class covariance. They showed that 1) higher $\mathcal{NC}_1$ in the pre-trained features led to a higher transfer accuracy and 2) lower  $\mathcal{NC}_1$ in the target-task features also led to a higher transfer accuracy.

**Summary Of The Review:**

Overall, I believe that the paper suffers from a somewhat sloppy experiment design, incomplete experiment description, and potentially misleading conclusions. The main conclusions are either already pointed out by other papers, or the direct consequence of those findings.

---

### Official Review · Reviewer_ny4r · 2022-10-23

**Confidence:** 4
**Correctness:** 3
**Technical Novelty And Significance:** 3
**Empirical Novelty And Significance:** 4
**Recommendation:** 5

**Clarity, Quality, Novelty And Reproducibility:**

Clarity
=====
The paper is clear and easy to follow for the most part.  One exception is the following.

In Figure 5 (left), the red stars look like a parabola to me. It seems that NC_1 increases as we go deeper, up to a ‘turning point’ after which it decreases as we go even deeper. This seems to contradict the discussion in the text that the smaller the NC_1, the better the transfer accuracy (since this isn’t the case for all parts of the parabola). This was confusing to me and it would be useful if the authors could comment on this.

Novelty
======
As far as I can tell, this investigation is novel.

Reproducibility
============
The proposed method, experiments and metrics are simple enough, and enough detail is provided that I believe I could reproduce them.

Quality, detailed comments, and more open-ended questions
===============================================
Most results in the paper show NC_1 numbers mostly, why not show DCNV numbers too?

If transferring between CIFAR-100 and CIFAR-10 it is perhaps expected that collapse in intra-class variance will lead to poor transfer, since those two datasets share (some of the) label space. It would be more interesting to investigate scenarios of less related source and target datasets. For example, it would be interesting to use the more diverse transfer datasets from section 3.2 for the experiments of section 3.1 too.

I wonder how the conclusions throughout the paper are affected by the (unusual) choice of freezing the feature extractor and only learning a linear layer during transfer. The alternative of allowing to finetune the feature extractor usually performs better, even for little or moderate sizes of training data from the target domain, so studying that setup would be more relevant (closer to state-of-the-art).

Head2toe [1] can also be seen as a sweet-spot between linear probing and finetuning: it trains a linear layer on top of feature from several layers (instead of only on top of penultimate layer features). What is the relationship between its performance and neural collapse?

It seems that there are 2 orthogonal approaches discovered for increasing transfer accuracy: one that influences pretraining (namely, adding projection layers even in supervised training can be helpful, increasing NC_1), and one that influences finetuning (namely, finetuning the penultimate layer too while training the downstream classifier, decreasing NC_1). Presumably the two are complementary and can be combined to achieve greater gains. Have the authors explored this?

References
==========
[1] Head2Toe: Utilizing Intermediate Representations for Better Transfer Learning. Evci et al. ICML 2022.

Minor comments
=============
Incorrect usage of citation style throughout the paper. Use \citep when referring to a paper and \citet when referring to its authors. For example: A) “Previous work (X et al) addressed the problem of …” and B) “X et al. (2021) pointed out that …” - the former would use \citep while the latter \citet.

‘the lose of information’ → ‘the loss of information’

‘are more/less collapse’ → ‘are more/less collapsed’ (there were a few occurrences of this)

‘Without otherwise specified’ → ‘unless otherwise specified’


**Strength And Weaknesses:**

Strengths
========
[+] The study of the relationship between neural collapse and transfer learning performance is novel to the best of my knowledge, interesting, and relevant, as it can offer an (additional) explanation for why increasing diversity (e.g. with data augmentation) and projection layers can boost transfer performance, for example.

[+] The paper is well-written, clear and easy to understand for the most part.

[+] The experiment design is technically sound.

Weaknesses
==========
[-] The main weakness I see is the limited settings considered which, due to being an empirical paper, make me question the generality of the conclusions. Most prominently, the choice to learn a linear layer on top of a frozen network at transfer time, and the limited choice of datasets to transfer to (see detailed comments below).

[-] In terms of the relationship between NC in pretrained representations and transfer accuracy, the story seems to be quite nuanced. I didn’t gain an understanding about when we should expect the two to be positively versus negatively correlated. What happens at the ‘turning point’ in Figure 2?  Is there some method that can be used to find the ‘sweet spot’ in terms of NC_1 that is just right for achieving best transfer performance? The investigation there feels a bit premature in the sense that there are still many open questions. Partially because of this, it’s hard to walk away with a solid take-home message from the paper.


**Summary Of The Paper:**

This paper investigates the neural collapse phenomenon (minimal intra-class variance and maximally separated class means in learned representations of neural networks) in the context of transfer learning. They find that this type of collapse in the representations of the pretrained model hurts transfer performance (though this is only true up to a threshold), but neural collapse in the representations obtained after training on the target task actually correlates with better transfer performance. Their intuition is that capturing all the intra-class variation ‘upstream’ leads to more diverse representations that are likely to transfer well, whereas collapsing in this way ‘downstream’ is a sign of fitting the target task well which increases performance. They also ran a set of interesting experiments investigating the collapse in intermediate layers of the deep network, and how that correlates with transfer performance obtained by training a classifier on top of the corresponding intermediate representation. Their insights led them to hypothesize that finetuning not only a linear classifier but also the penultimate layer during transfer is helpful and indeed increases neural collapse as well as downstream accuracy.

**Summary Of The Review:**

Overall, I found the paper well-written, the topic explored interesting and relevant and the investigation novel, to the best of my knowledge. Some findings were interesting and not previously known. However, the empirical exploration is limited in some important ways and I’m wondering whether the findings would generalize to different settings (see above). It’s also hard to take-away a clear message from the paper due to the premature nature of the investigation in some cases (see above discussion about the relationship between NC in pretrained representations and transfer performance). I therefore recommend only a weak acceptance of the paper in its current form.

---

### Official Review · Reviewer_ETik · 2022-10-26

**Confidence:** 4
**Correctness:** 3
**Technical Novelty And Significance:** 3
**Empirical Novelty And Significance:** 2
**Recommendation:** 3

**Clarity, Quality, Novelty And Reproducibility:**

The manuscript is overall clearly written. The authors provide the code in the supplements, so I believe the results are reproducible.

**Strength And Weaknesses:**

The paper is overall well-written and easy to follow. Relating NC with transferability is interesting. NC also provides a perspective to understand how various factors affect transferability.

Weaknesses:

1.There is no evidence showing that the relationship between NC and transferability is robust. As the authors already mentioned, a large NC leads might not lead to good transfer performance as well, e.g. the model is randomly initialized and not trained. This is a simple sanity check that the correlation between NC and transferability does not pass.

2.It is unclear what it the causality between NC, transferability, and the diversity of features. To me, learning diverse features in pre-training is the cause, while NC and transferability are consequences. Therefore, using NC to understand transferability is misleading in this sense. The idea that learning less diverse features leads to bad transfer performance is not novel. [1] contains ideas like this. Self-supervised learning contains more transferable features than supervised learning, so that it is more transferable.

3.It is also not surprising at all that NC on the downstream tasks correlated well with the performance on downstream tasks. Even without pre-training, this correlation should be true, because small NC means the margin in the classification problem is large. [5] already showed that this is the case for few-shot learning.

4.Lacking comparison with other works on transferability. There exist a line of works on predicting the transferability of the models with various metric. See [2] and references therein. The authors should provide a comparison with them to give the readers a sense how NC perform as a metric of transferability.

5.Lacking justifications on larger datasets. It’s better to provide the NC results on ImageNet apart from CIFAR100 and CIFAR-10. Evaluating the numbers based on ImageNet should not be difficult with the publicly available pre-trained models? This gives readers more confidence on this phenomenon.

6.The proposed transfer algorithm is not novel. Training more than one layers should definitely perform better than linear probe. The authors should also provide the numbers of fine-tuning as a comparison. Besides, there are a bunch of works on efficient fine-tuning, such as fine-tuning only the bias [3], and adapters [4]. It would be better to compare with them as well.

Minors:

Abstract, “when pretrain models”, Should be pre-training

Basics of NNs, Is the layer index notation used anywhere else? If not, including it here will only make it more cluttered. Besides, for resnets, it is not correct definition. The definition here only applies to MLPs.

[1] Self-supervised Learning is More Robust to Dataset Imbalance.
[2] LogME: Practical Assessment of Pre-trained Models for Transfer Learning.
[3] BitFit: Simple Parameter-efficient Fine-tuning for Transformer-based Masked Language-models.
[4] VL-Adapter: Parameter-Efficient Transfer Learning for Vision-and-Language Tasks.
[5] Unraveling meta-learning: understanding feature representations for few-shot tasks.


**Summary Of The Paper:**

This paper studies the relationship between transferability and neural collapse (NC). The authors find out that the NC can be a double-bladed sword in transfer learning. For pre-training, larger NC indicates the model learns richer features from pre-training, leading to better transfer performance. For downstream tasks, smaller NC indicates the classes are better separated, and the model potentially generalizes better on downstream tasks. The authors further propose tuning one additional layer to improve over linear probe.

**Summary Of The Review:**

Although I like the topic of this paper, I don’t think it fulfill its goal adequately and meet the standard of ICLR as a top venue. Thus, I tend to reject this manuscript based on the above weaknesses.

---

### Official Review · Reviewer_urV1 · 2022-10-28

**Confidence:** 4
**Correctness:** 3
**Technical Novelty And Significance:** 3
**Empirical Novelty And Significance:** 3
**Recommendation:** 5

**Clarity, Quality, Novelty And Reproducibility:**

Overall, the paper is easy to read. The findings are interesting and potentially useful for transfer learning. The experiments are thorough, although being rather small scale and for classification tasks only.  The source code is provided, but not checked by the reviewer.

**Strength And Weaknesses:**

Strength:

1) The empirical study is comprehensive for classification tasks.
2) The aforementioned two findings are interesting.


Weaknesses:

1) The presented empirical study mainly focuses on classification tasks mostly with rather small models (ResNet 18/50) and datasets which share similar nature between pretrain and downstream (e.g., CIFAR100/MiniImageNet  to CIFAR10).  The study on VIT-B is only test on CIFAR100. It will be interesting to see if the findings will retain between more diverse datasets such as the Visual Domain Decathlon dataset, or even from classification tasks to object detection and segmentation tasks.

2) The performance on the pretrain data are not presented (e.g. in Table 1 and Fig. 2). Based on the findings, larger NC metrics on the pretrained data seem to indicate inferior performance on the pretrained data.  It may worth considering the performance on the source dataset, the NC, and the performance on the downstream datasets jointly.

**Summary Of The Paper:**

This paper presents an empirical study on the relationships between the neural collapse (NC) and the transferability of DNNs from source datasets to downstream tasks. The class-distance normalized variance (CDNV) is used as the metric of evaluating NC. The empirical study is done using ResNet18/50 (from CIFAR100 or MiniImageNet to CIFAR10) and ViT-B (from JFT & ImageNet22k to CIFAR100). Several findings are obtained: (1) If the learned representations are less collapse on the pre-trained data (i.e. larger NC metrics), they often lead to better transfer accuracy with the intuitive explanations that they better preserve the intrinsic structures of the input data.  This positive relationship only holds up to certain threshold with the intuitive explanations that the increase of the NC metrics (i.e., larger feature variability) will significantly decrease the margin upon certain threshold. (2) On the downstream tasks themselves, the relationships are negatively preserved, that is the more collapsed the representations of the layer on the downstream data, the better the transfer accuracy.

**Summary Of The Review:**

Overall, the paper is a good paper, providing interesting insights on the relationships between neural collapse and the transferability.  If the empirical study is done with larger scale and the findings retain, the paper will be much more interesting to the community.

---

### Author Response · Authors · 2022-11-18
**Revision Summary**

We thank all the reviewers for their detailed and thoughtful comments. We are encouraged that all the reviews found our result interesting and the manuscript easy to follow. We noticed that some reviewers pointed out that our experiments are not comprehensive enough in terms of limited datasets and network architectures, and others point out that our proposed fine-tuning method is a bit simple with limited comparison. Based on suggestions from all the reviewers, we have included more comprehensive empirical results with a new fine-tuning method inspired by our findings, which we revised in Section 3.3. To summarize, we have carefully revised the manuscript in these aspects:
1. We add MSE and SupCon loss results to Figure 1 to support our claims that the success of projection heads cannot be entirely attributed to the contrastive-based loss, rather, the benefit comes from the layer-wise progressive collapse nature of neural networks.
2. To make the above argument more sound, in Figure 9, we further reproduced the result in [1] which states that the log of separation metric (NC1) has a universal (linearly) decay pattern after training across network architectures and datasets.
3. Based on our findings, we redesigned our fine-tuning method which requires only training one additional layer plus a skip connection term for linking the fine-tuned layer features with the penultimate layer features. We demonstrate the effectiveness of the method using pre-trained ResNets and Vit-B32 models on various downstream datasets.

[1] A Law of Data Separation in Deep Learning

---

### Decision · Program_Chairs · 2023-01-20

**Decision:**

Reject

**Justification For Why Not Higher Score:**

While the paper studies a very interesting problem, the current study is superficial to some extent and the observations are not robust to different settings. In addition, results on large datasets and more sophisticated transfer methods requested by the reviewers were not provided. Reviewers complained about the ineffective rebuttal and converged to unanimous rejection. AC finds no evidence to bump up.

**Justification For Why Not Lower Score:**

N/A

**Metareview: Summary, Strengths And Weaknesses:**

This paper studies a very interesting yet important problem: how does mode collapse impact transferability in deep representation learning. Some metrics are defined to enable a quantitative study of the relationship, which are are good research cycle. Reviewers are generally in favor of the topic studied, but are not satisfied with the superficial study presented in the current paper: The empirical study is mainly carried out on small datasets and classification tasks using simple transfer learning methods; The observations of the relationship are not generally robust and cannot shed enough lights for future practices; just to name a few. Authors did not provide an effective rebuttal with new results and evidences requested by the reviewers. As a result, most reviewers confirmed that their comments were not well addressed, and unanimously recommended rejection. AC has to reject the paper in its current form based on the above reasons, but feels that the paper is promising for another venue after a major revision that addresses the comments.